# A conserved megaprotein-based molecular bridge critical for lipid trafficking and cold resilience

Changnan Wang[1,11], Bingying Wang[1,11], Taruna Pandey[1], Yong Long[2], Jianxiu Zhang [3], Fiona Oh[1], Jessica Sima[1], Ruyin Guo[1], Yun Liu[4], Chao Zhang [5], Shaeri Mukherjee[6], Michael Bassik [7], Weichun Lin[4], Huichao Deng [8], Goncalo Vale [9], Jeffrey G. McDonald [9], Kang Shen [8] & Dengke K. Ma [1,10] ✉

Cells adapt to cold by increasing levels of unsaturated phospholipids and membrane fluidity through conserved homeostatic mechanisms. Here we report an exceptionally large and evolutionarily conserved protein LPD-3 in *C. elegans* that mediates lipid trafficking to confer cold resilience. We identify *lpd-3* mutants in a mutagenesis screen for genetic suppressors of the lipid desaturase FAT-7. LPD-3 bridges the endoplasmic reticulum (ER) and plasma membranes (PM), forming a structurally predicted hydrophobic tunnel for lipid trafficking. *lpd-3* mutants exhibit abnormal phospholipid distribution, diminished FAT-7 abundance, organismic vulnerability to cold, and are rescued by Lecithin comprising unsaturated phospholipids. Deficient *lpd-3* homologues in Zebrafish and mammalian cells cause defects similar to those observed in *C. elegans*. As mutations in *BLTP1*, the human orthologue of *lpd-3*, cause Alkuraya-Kucinskas syndrome, LPD-3 family proteins may serve as evolutionarily conserved highway bridges critical for ER-associated non-vesicular lipid trafficking and resilience to cold stress in eukaryotic cells.

Homeoviscous adaptation (HVA) refers to the ability of cells to adjust membrane viscosity by changing cell membrane lipid compositions and unsaturation in response to environmental temperature shifts[1–3]. For example, exposure to cold temperature in bacteria rigidifies cell membrane, triggering HVA to maintain membrane fluidity within a normal range to ensure proper activity of membrane proteins[2]. Besides bacteria, HVA has been observed in many eukaryotic organisms as an evolutionarily conserved mechanism that enables adaptation to changes in environmental temperature. In both bacteria and the

multicellular model organism *C. elegans*, HVA involves temperature-triggered transcriptional regulation of genes encoding lipid desaturases. While heat down-regulates a fatty acid desaturase-encoding gene *fat-7* through *acdh-11*, cold up-regulates *fat-7* through the membrane fluidity sensor PAQR-2 and downstream transcriptional regulators in *C. elegans*[3–6]. Temperature-regulated FAT-7 catalyzes chemical C=C double bond formation in fatty acyl chains, leading to membrane lipid unsaturation and increased membrane fluidity. HVA through such regulation of lipid desaturases facilitates cellular

[1]Cardiovascular Research Institute and Department of Physiology, University of California San Francisco, San Francisco, CA, USA. [2]State Key Laboratory of Freshwater Ecology and Biotechnology, Institute of Hydrobiology, Chinese Academy of Sciences, Wuhan, China. [3]Department of Molecular and Cellular Physiology, Stanford University, Stanford, CA, USA. [4]Department of Neuroscience, University of Texas Southwestern Medical Center, Dallas, TX, USA. [5]Shanghai Institute of Precision Medicine, Shanghai Ninth People's Hospital, Shanghai Jiao Tong University School of Medicine, Shanghai, China. [6]Department of Microbiology and Immunology, University of California, San Francisco, San Francisco, CA, USA. [7]Department of Genetics, Stanford University School of Medicine, Stanford, CA, USA. [8]Department of Biology, Howard Hughes Medical Institute, Stanford University, Stanford, CA, USA. [9]Center for Human Nutrition, University of Texas Southwestern Medical Center, Dallas, TX, USA. [10]Innovative Genomics Institute, University of California, Berkeley, CA, USA. [11]These authors contributed equally: Changnan Wang, Bingying Wang. ✉e-mail: Dengke.Ma@ucsf.edu

adaptation to, and orgasmic survival against, environmental temperature stresses[1–3].

In eukaryotes, lipid biosynthetic enzymes and lipid desaturases, including FAT-7, are located at the endoplasmic reticulum (ER). The newly synthesized and unsaturated lipids can distribute to other cellular organelles by both well-characterized vesicular transport pathways and less well-understood non-vesicular transport mechanisms[7,8]. Earlier studies indicate that inhibition of vesicular transport pathways does not substantially decrease transfer of phospholipids, including phosphatidylcholine (PC) and phosphoatidylethanolamine (PE), from ER to plasma membranes (PM)[9,10]. More recent studies suggest that non-vesicular lipid trafficking among various intracellular organelles, including ER, lysosomes and mitochondria, can occur through a conserved family of RBG domain-containing VPS13-like non-vesicular lipid transporters[11–16]. However, compared to the vesicular lipid transport pathways, the mechanisms of action, physiological regulation and organismic functions of non-vesicular lipid transporters remain still largely unknown.

We performed a mutagenesis screen for genetic suppressors of FAT-7 in *C. elegans* and identified *lpd-3*, which encodes a 452 kDa megaprotein bridging the ER and PM. AlphaFold2-assisted structural prediction reveals an elongated hydrophobic tunnel in LPD-3 suited for lipid trafficking. We show that LPD-3 is critical for *fat-7* expression, normal distribution of phospholipids at the PM, and organismic resilience to severe cold stress. Mutations in *KIAA1109/BLTP1*, the human orthologue of *lpd-3*, cause an autosomal recessive disorder, Alkuraya-Kucinsk syndrome[17–20]. We found that decreased expression of *lpd-3* homologues in Zebrafish and mammalian cells elicited similar phenotypes as in *C. elegans*. Our results suggest evolutionarily conserved roles of the LPD-3 family proteins as megaprotein-based molecular bridges in non-vesicular trafficking of lipids and stress resilience to cold temperature.

## Results

### Genetic screens identify LPD-3 as a critical regulator of FAT-7

We have previously discovered components of a genetic pathway in *C. elegans* that maintains cell membrane fluidity by regulating lipid unsaturation via the fatty acid desaturase FAT-7 in response to temperature shifts[3]. Loss-of-function mutations in the gene *acdh-11* cause constitutive FAT-7 up-regulation. In forward genetic screens to isolate mutants with *acdh-11*-like constitutive expression of *fat-7*::GFP, we identified several alleles of *acdh-11* and two additional genes, *cka-1* and *sams-1* (Fig. 1a, Supplementary Fig. 1a, b), which are involved in cellular phosphatidylcholine biosynthesis[21–23]. *acdh-11*, *cka-1* and *sams-1* encode negative regulators of *fat-7*. To identify positive regulators of *fat-7*, we performed *acdh-11* suppressor screens for mutants with diminished *fat-7*::GFP (Fig. 1a). Unlike loss-of-function mutants of known positive regulators (e.g., *nhr-49/80* or *sbp-1* with complete loss of *fat-7*::GFP signals)[24–27], a rare *acdh-11* suppressor mutant *dma544* exhibits diminished *fat-7*::GFP in the anterior intestine and decreased (but still visible) *fat-7*::GFP in the posterior intestine (Supplementary Fig. 2a, b). By single nucleotide polymorphisms-based genetic mapping and whole-genome sequencing, we identified *dma544* as a missense transition mutation of the gene *lpd-3*. RNAi against *lpd-3*, an independently derived deletion mutation or another *acdh-11* suppressor *dma533* recapitulated both *fat-7*::GFP suppression and the morphological pale phenotype of *dma544* (Fig. 1b, c, f). RNAi against *lpd-3* also suppressed *fat-7*::GFP in the *cka-1* or *sams-1* mutants (Supplementary Fig. 1c).

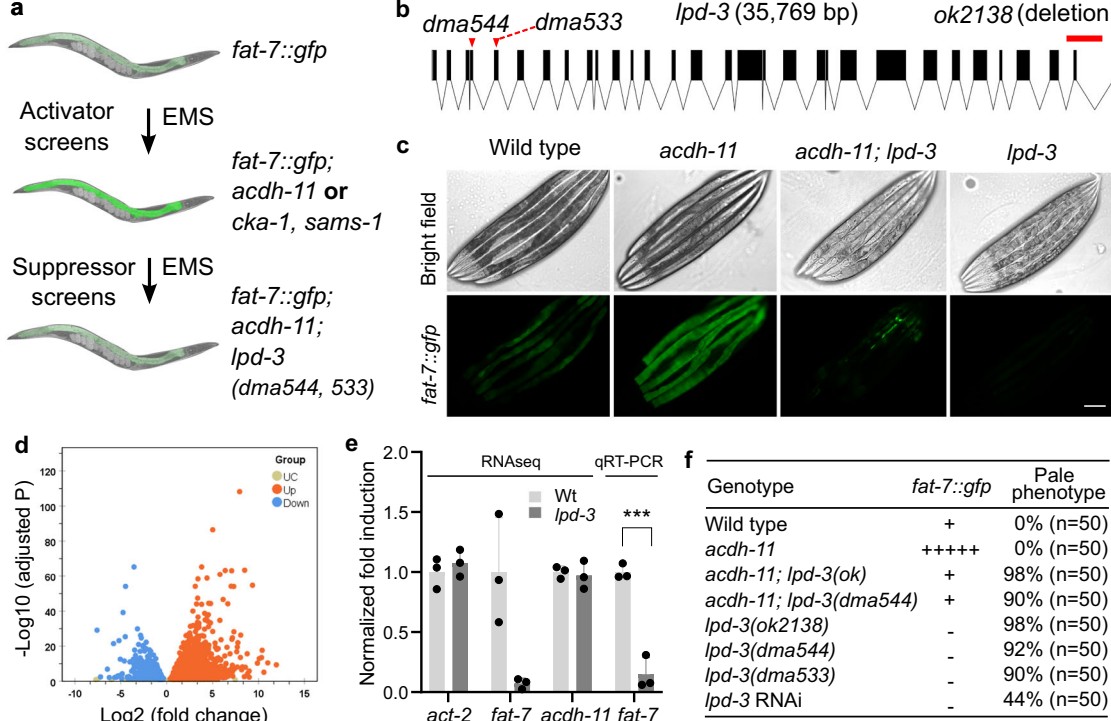

**Fig. 1 | Genetic identification and transcriptomic analysis of *lpd-3*. a** Schematic of genetic screens that led to the identification of *acdh-11*, *cka-1*, and *sams-1* as negative regulators of *fat-7* and *acdh-11*-suppressing *lpd-3* as positive regulator of *fat-7*. **b** Full-length gene diagram of *lpd-3* with the point mutations *dma544*, *dma533* (arrows) and deletion mutation *ok2138* (line). **c** Representative images of *fat-7*::GFP animals in wild type, *acdh-11(n5857)* single, *acdh-11(n5857); lpd-3(ok2138)* double, or *lpd-3(ok2138)* single mutants. Scale bar, 50 μm. **d** Volcano plot showing significantly (adjusted *p* value < 0.05, Wald test in DESeq2) up- (orange) or down- (blue) regulated genes in *lpd-3* mutants compared with wild type. **e** Normalized fold induction (RNAseq and qRT-PCR) of *fat-7* showing its diminished expression in *lpd-3* mutants. Values are means ± S.D. ***P* < 0.001 (*N* = 3 biological replicates, *P* = 0.0006 from two-sided unpaired *t*-test). **f** Table summary of *fat-7*::GFP abundance (indicated by relative numbers of plus signs based on fluorescent intensities or minus sign, no signals) and the morphological pale phenotypes in animals with indicated genotypes (single, double mutations or RNAi) and phenotypic penetrance. Source data are provided as a Source Data file.

## LPD-3-regulated transcriptome and FAT-7-related phenotypes

To better understand functions of LPD-3, we assessed how *lpd-3* mutations alone might impact gene expression changes and *fat-7*-related phenotypes without *acdh-11* mutations. In the wild type, *fat-7*::GFP was increased upon exposure to a cold temperature at 15 °C, yet such increase was abolished in *lpd-3* mutants (Supplementary Fig. 2c, d). The baseline expression of *fat-7*::GFP at 20 °C was also abolished in *lpd-3* mutants, including that in the posterior intestine (Fig. 1c). We performed RNA sequencing (RNAseq) to compare transcriptomes of wild type versus *lpd-3* mutants cultivated at 20 °C. After differential expression analyses of triplicate samples, we identified 6251 genes that are significantly up- or down-regulated in *lpd-3* mutants (Fig. 1d, Supplementary data file 1). As expected, *fat-7* was one of the most highly down-regulated genes (log$_2$fold change = −5.05, adjusted *p* value = 2.54E−13), while expression of its upstream regulators including *cka-1*, *sams-1* or *acdh-11* remained largely unchanged (Fig. 1e, Supplementary data file 1). Among the genes that were significantly up-regulated (adjusted *p* value > 0.05), 234 genes were also up-regulated by exposure to 4 °C cold-warming stress[28], including the previously validated cold-inducible gene *asp-17* that we confirmed to increase dramatically in *lpd-3* mutants without cold exposure (Supplementary Fig. 3a–c). We used WormExp[29] to compare these 234 genes to expression data from all previously characterized mutant animals and found they were significantly similar (*P* = 1.7e−107) to the gene set regulated by RNAi against *sbp-1* (Supplementary Fig. 3d)[30]. RNAi against *lpd-3* or *sbp-1* has previously been shown to induce the morphological pale and lipid depletion phenotypes[25,31–33]. We confirmed such phenotype in *lpd-3(dma544)* mutants (Fig. 1c, f) and further showed that deletion of LPD-3 caused markedly fewer and smaller lipid droplets using an established lipid droplet marker DHS-3::GFP[34] (Supplementary Fig. 3e). We made similar observation in animals with RNAi against *sbp-1*, which encodes a master regulator of lipogenesis and *fat-7* expression for unsaturated lipid accumulation in *C. elegans*[24,25,35,36] (Supplementary Fig. 3e). These results identify transcriptomic changes as well as *fat-7*-related lipid and morphological phenotypes in *lpd-3* mutants.

## Structural features and cellular localizations of LPD-3

We next examined structural features of LPD-3 that may provide insights into its molecular function. LPD-3 is an exceptionally large protein, consisting of predicted 4018 amino acid residues of 452 kDa molecular weight. We sought to obtain a predicted LPD-3 structure by the machine-learning-based AlphaFold2 program[37]. As the program is limited to polypeptides smaller than 2500 amino acids, we segmented the full-length LPD-3 sequence into four overlapping parts that were separately predicted and then rejoined to generate a full-length structure (Fig. 2a). The yielded full-length structure reveals an ~30 nm-long rod-like shape consisting of twisted β-sheets that form a striking tubular cavity and internal hydrophobic tunnel extending along its entire length (Fig. 2a, b, Supplementary Data file 2). The *dma544* mutation (G200E) disrupts a highly conserved glycine residue lining up the internal tubular wall while the resulting G200E glutamic acid residue of LPD-3 is predicted to partially block the tunnel entry (Fig. 2c). The N-terminal sequence of LPD-3 forms a putatively hydrophobic transmembrane helix while its C-terminal sequence harbors an amphiphilic patch and a polybasic cluster (KxKK motif that binds to PIP2/PIP3) indicative of association with the cytosolic side of lipid membranes by electrostatic interaction[38] (Fig. 2d). These structural features of LPD-3 are reminiscent of the recently described VPS13/ATG2 family transporters that mediate non-vesicular lipid trafficking across organelle membranes[11–13,16], although these separate families of proteins lack apparent protein sequence similarities.

To determine the subcellular localization of LPD-3, we constructed transgenic reporters for both N- and C-termini of LPD-3. A mCherry-tagged N-terminal LPD-3 translational reporter showed prominently discrete intracellular signals in the intestine (Fig. 2e). We crossed this N-terminal reporter into established *C. elegans* strains expressing bright fluorescent GFP directed to various intracellular organelles including Golgi (mans::GFP), mitochondria (MAI-2::GFP), peroxisome (GFP::DAF-22), lysosome (LMP-1::GFP), endosome (RAB-7::GFP) and ER membranes (GFP::C34B2.10::SP12). We found that the N-terminal LPD-3Nt::mCherry signals were closely surrounded by ER membrane markers (Fig. 2e). By contrast, a mCherry-tagged C-terminal LPD-3 translational reporter displayed both intracellular and plasma membrane (PM) signals in the intestine (Fig. 2f). We found that the PM signal of mCherry-tagged C-terminal LPD-3 co-localized with Akt-PH::GFP, an established reporter that binds to the phospholipid PIP3 of the inner leaflet of PM[39] (Fig. 2f). We also used CRISPR/Cas9 to generate knock-in of seven copies of GFP11 at the C-terminus of endogenous LPD-3 and reconstitution with GFP1-10 revealed that endogenous LPD-3::GFP localized to ER and co-localized with mScarlet-labeled EYST-2 and MAPPER, specific markers for ER-PM junctions[40–42], but not with a Golgi marker (Fig. 2g, h and Supplementary Fig. 4). Thus, LPD-3 primarily localizes to the ER, particularly at the ER-PM membrane contact sites that are known to mediate lipid trafficking and integrate phospholipid regulation[43–45].

## Essential roles of LPD-3 in ER-to-PM lipid trafficking and SBP-1 regulation

The structural features and cellular localizations of LPD-3 indicate an ER-to-PM bridge-like tunnel with plausible roles in mediating non-vesicular ER-to-PM trafficking of lipids. Next, we conducted a series of functional assays and phenotypic analyses to test this idea.

First, we examined how LPD-3 may impact phospholipid distribution in the cell. Phospholipids, including phosphatidylcholine (PC), phosphatidylserine (PS), phosphatidylethanolamine (PE) and phosphatidylinositol (PI), are newly synthesized at the ER and transported to cytoplasmic membranes through both vesicular and non-vesicular mechanisms[7,8]. Since probes for live monitoring of most phospholipid distribution are unavailable in *C. elegans*, we took advantage of the genetically encoded reporter Akt-PH::GFP, which binds to the phospholipid PIP3 (3,4,5-phosphate), to assess the intracellular distribution and abundance of PIP3 species[39]. We found that wild-type animals exhibited Akt-PH::GFP fluorescence enriched along the apical membrane of the intestine (Fig. 3a). By contrast, when crossed into *lpd-3* mutants, this same reporter at the same developmental stage (24 or 48 h after L4) showed attenuated overall fluorescence without apparent apical enrichment in the intestine, and more dispersed intracellular distribution compared to that in wild type (Fig. 3a). We also noticed that transgenic expression of mCherry-tagged C-terminal LPD-3 reduced the apical enrichment of Akt-PH::GFP, indicating competition of both reporters for the same substrate (Fig. 2f). As PM-localized PIP2/PIP3 is associated with and often stimulates actin polymerization, we found that a filamentary actin reporter *act-5*::GFP in the intestine[46] also displayed markedly reduced abundance and apical localization in *lpd-3* mutants (Fig. 3b). These results reveal striking defects of PIP3-binding reporter distribution in LPD-3-deficient intestinal cells and support the notion that LPD-3 normally promotes enrichment of phospholipids, at least certain PI species or their precursors, at the PM.

Second, we examined how LPD-3 may impact functional consequences of loss of SAMS-1. The S-adenosyl methionine synthetase SAMS-1 is critical for the biosynthesis of phosphatidylcholine in *C. elegans*, decreased abundance of which in ER membranes activates ER stress response and expression of lipogenic genes including *fat-7* via SBP-1 regulation[21,47]. We found that *fat-7*::GFP was strongly activated by RNAi against *sams-1* in wild type but not in *lpd-3* mutants (Fig. 3c, d). We made similar observation on *hsp-4*p::GFP, an established reporter for ER stress response (Fig. 3e). These results indicate that LPD-3 antagonizes effects of SAMS-1 in PC accumulation in ER membranes, again supporting a physiological role of LPD-3 in facilitating the ER-to-

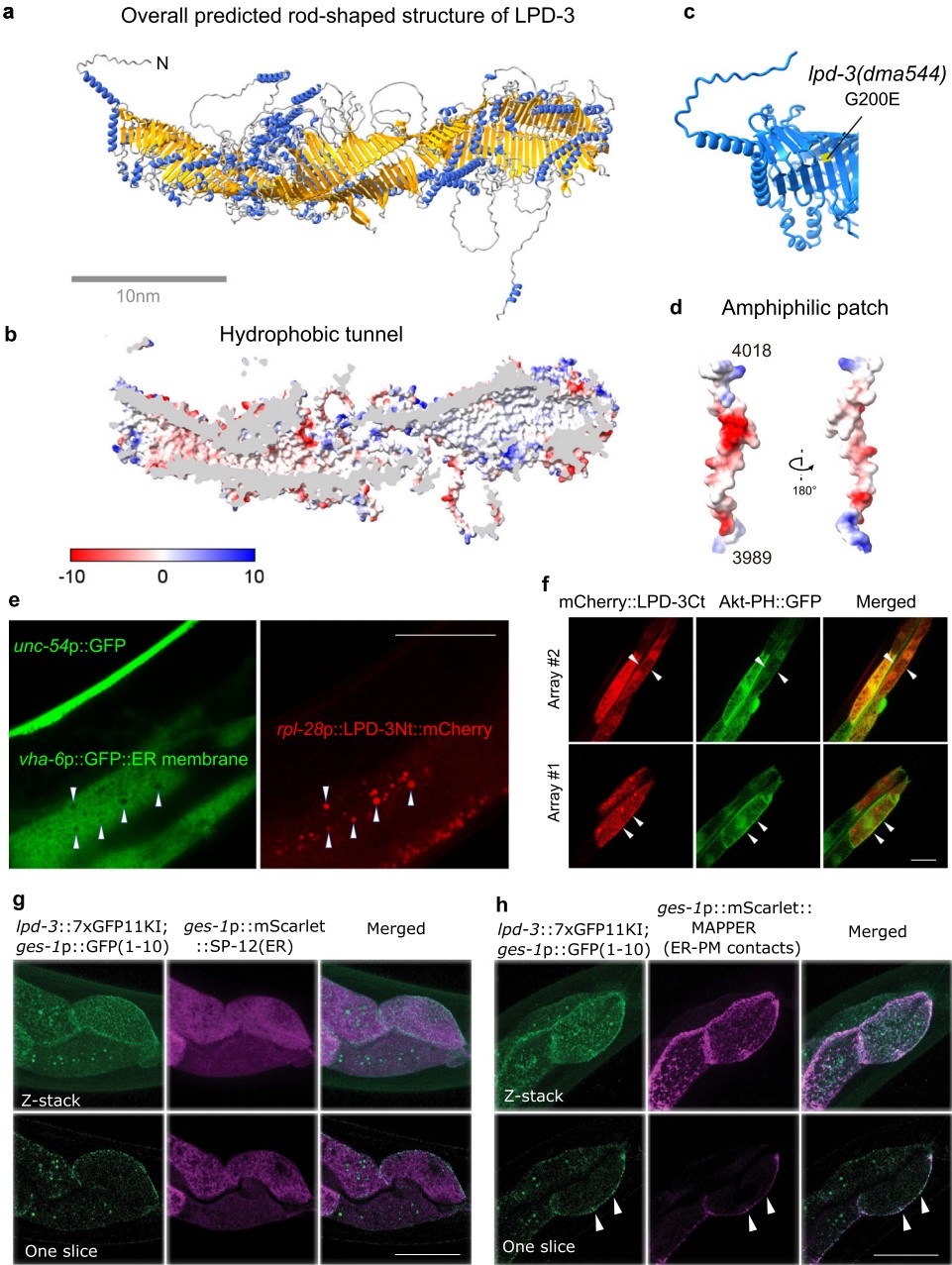

**Fig. 2 | Structural features and cellular localizations of LPD-3. a** Overall structure of the full-length LPD-3 assembled from four segments whose structures were separately predicted by AlphaFold v2.0, with both N and C-termini noted. **b** Cross-sectional view of the LPD-3 structure showing the hydrophobic tunnel running continuously along the entire inside. **c** Ribbon representation of the N-terminal part of LPD-3, with the G200E mutation indicated in yellow. **d** Structure of the LPD-3 C-terminus showing an amphiphilic patch (hydrophobic, red; hydrophilic, blue) that was used to generate mCherry-fused LPD-3 reporters. **e** Representative fluorescence images showing co-localization (arrow heads) of ER membrane markers with a mCherry fusion reporter of the LPD-3 N-terminus (*rpl-28*p::LPD-3Nt::mCherry with LPD-3 a.a. 1-72, *unc-54*p::GFP as co-injection marker). **f** Representative fluorescence images (from two independent transgenic extrachromosomal arrays #1 and #2) showing co-localization (arrow heads) of Akt-PH::GFP that binds to PM-PIP3 with a mCherry fusion reporter (*ges-1*p::mCherry::LPD-3Ct, a.a. 3945-4018) of the LPD-3 C-terminus. **g** Representative confocal fluorescence images showing endogenous LPD-3::GFP (generated by CRISPR/Cas9-mediated knock-in and split GFP complementation) in apposition with mScarlet::SP-12(ER). **h** Representative confocal fluorescence images showing endogenous LPD-3::GFP that co-localizes (arrow heads) with mScarlet::MAPPER, a marker for ER-PM junctions. Scale bar, 50 μm.

PM trafficking of phospholipids, which reduces PC accumulation in ER membranes.

Third, we examined how LPD-3 may impact the nuclear abundance of SBP-1, a master regulator of lipogenesis and *fat-7*. The *C. elegans* SREBP homolog SBP-1 promotes lipogenesis and transcriptionally activates *fat-7* expression by translocating from ER membranes to nucleus[21,24,25]. We found that the abundance of nuclear SBP-1::GFP was markedly decreased by RNAi against *lpd-3* (Fig. 3f). By contrast, a transcriptional *sbp-1*p::GFP reporter was not apparently affected by *lpd-3* RNAi (Fig. 3g). These results indicate that LPD-3 promotes *fat-7* expression likely through post-transcriptional regulation of SBP-1. Since low PC levels in ER membranes trigger SBP-1 nuclear translocation, these data are consistent with the notion that LPD-3 decreases PC levels in ER membranes by promoting ER-to-PM phospholipid trafficking.

Fourth, we examined how LPD-3 may impact cellular membrane integrity. Phospholipids with proper compositions of saturated and unsaturated fatty acyl chains are critical for the maintenance of

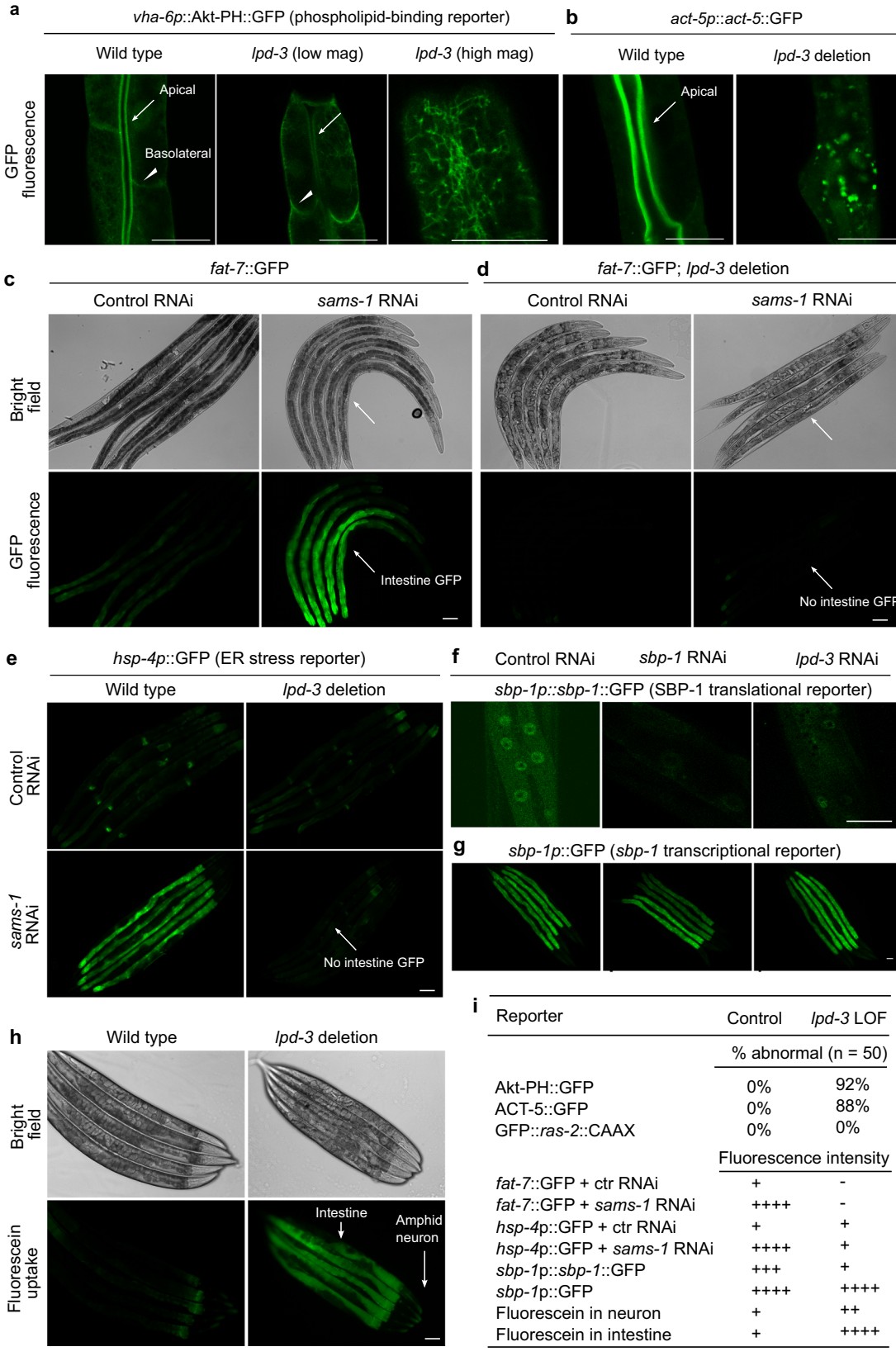

**i**

| Reporter | Control | lpd-3 LOF |
|---|---|---|
| | % abnormal (n = 50) | |
| Akt-PH::GFP | 0% | 92% |
| ACT-5::GFP | 0% | 88% |
| GFP::ras-2::CAAX | 0% | 0% |
| | Fluorescence intensity | |
| fat-7::GFP + ctr RNAi | + | - |
| fat-7::GFP + sams-1 RNAi | ++++ | - |
| hsp-4p::GFP + ctr RNAi | + | + |
| hsp-4p::GFP + sams-1 RNAi | ++++ | + |
| sbp-1p::sbp-1::GFP | +++ | + |
| sbp-1p::GFP | ++++ | ++++ |
| Fluorescein in neuron | + | ++ |
| Fluorescein in intestine | + | ++++ |

membrane fluidity and integrity. Using a fluorescein-based SMURF assay to measure membrane permeability[48], we found that *lpd-3* mutants accumulated markedly higher levels of fluorescein in the intestine, and to a lesser extent in amphid sensory neurons, compared to wild type (Fig. 3h, i). These results suggest that loss of LPD-3 may compromise intestinal PM integrity. As insufficient fatty acyl unsaturation of phospholipids causes membrane leakiness via formation of domains with high-order phases that lack plasticity[49], we predict that loss of LPD-3 may lead to retention of excessively unsaturated phospholipids in ER membranes. Indeed, we found that excessive lipid saturation in ER membranes by RNAi against *mdt-15* activated ER stress response[47] in wild type but not *lpd-3* mutants (Supplementary Fig. 5a).

**Fig. 3 | Essential roles of LPD-3 in ER-to-PM lipid trafficking and SBP-1 regulation. a** Representative confocal fluorescence images showing PIP3-binding Akt-PH::GFP reporters in wild type (apical intestinal membrane, arrow; basolateral membrane, arrow head) and *lpd-3(ok2138)* mutants at both low and high magnifications. **b** Representative confocal fluorescence images showing the actin reporter *act-5p::act-5::GFP* in wild type (apical intestinal membrane, arrow) and *lpd-3(ok2138)* mutants. **c** Representative bright-field and epifluorescence images showing activation of *fat-7*::GFP by RNAi against *sams-1*. **d** Representative bright-field and epifluorescence images showing activation of *fat-7*::GFP by RNAi against *sams-1* in wild type but not *lpd-3(ok2138)* mutants (arrow). **e** Representative epifluorescence images showing activation of the *hsp-4*p::GFP ER stress reporter by RNAi against *sams-1* in wild type but not *lpd-3(ok2138)* mutants (arrow). **f** Representative confocal fluorescence images showing reduced abundance of nuclear *sbp-1*p::*sbp-1*::GFP by RNAi against *sbp-1* or *lpd-3*. **g** Representative epifluorescence images showing no apparent changes of *sbp-1*p::GFP by RNAi against *sbp-1* or *lpd-3*. **h** Representative epifluorescence images showing markedly increased membrane permeability for fluorescein in *lpd-3(ok2138)* mutants. **i** Table summary of reporter phenotypes of indicated genotypes or conditions. Scale bars, 50 μm.

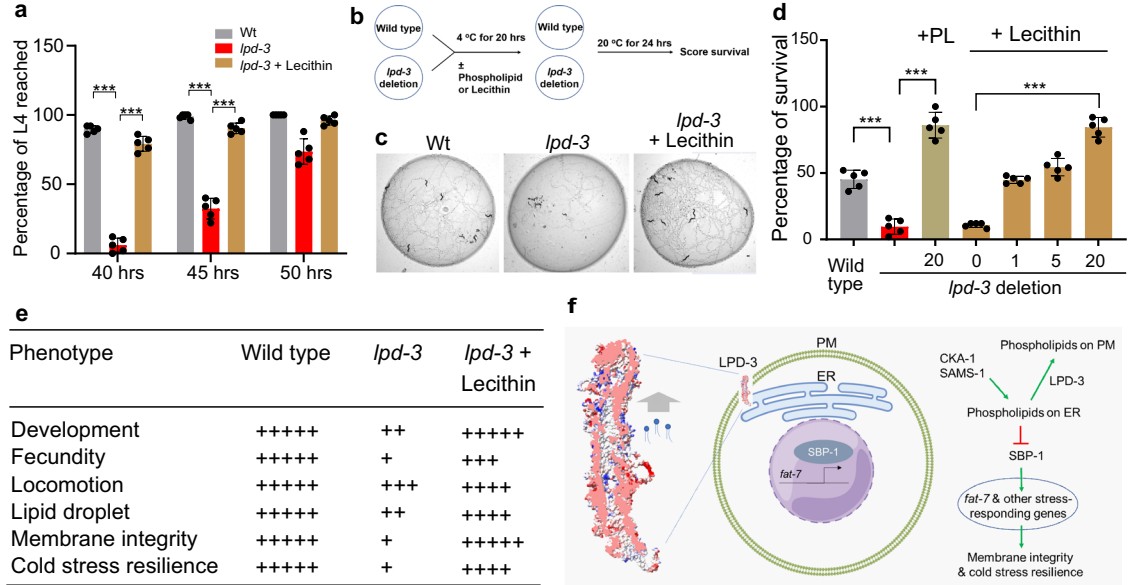

| Phenotype | Wild type | *lpd-3* | *lpd-3* + Lecithin |
|---|---|---|---|
| Development | +++++ | ++ | +++++ |
| Fecundity | +++++ | + | +++ |
| Locomotion | +++++ | +++ | ++++ |
| Lipid droplet | +++++ | ++ | ++++ |
| Membrane integrity | +++++ | + | +++++ |
| Cold stress resilience | +++++ | + | ++++ |

**Fig. 4 | Phospholipid/Lecithin rescues phenotypes of *C. elegans lpd-3* mutants. a** Percentages of animals that reached developmental L4 stage at indicated hours post egg preparation in wild type, *lpd-3(ok2138)* mutants with or without supplementation of Lecithin since egg preparation. Values are means ± S.D. ***P < 0.001 (two-way ANOVA followed by multiple two-sided unpaired *t*-tests, N = 5 biological replicates). **b** Schematic of the experiment to measure cold resilience of wild type and *lpd-3(ok2138)* mutants with supplementation of soy phospholipids or Lecithin for 6 hrs post developmental L4 stage. **c** Representative bright-field images showing markedly decreased survival of *lpd-3(ok2138)* mutants against cold exposure (4 °C for 20 h) and rescued survival by Lecithin supplementation. Scale bars, 100 μm. **d** Percentages of survived animals against cold exposure (4 °C for 20 h) with indicated genotypes and increasing doses of Lecithin or soy phospholipids (PL) that rescued *lpd-3* mutants. Values are means ± S.D. ***P < 0.001 (one-way ANOVA followed by multiple two-sided unpaired *t*-tests, N = 5 biological replicates). **e** Table summary of observed *lpd-3* phenotypes and their degrees of rescue (indicated by the numbers of + signs) by Lecithin (10 mg/ml on NGM). **f** Model to illustrate molecular functions of LPD-3 in mediating non-vesicular ER-to-PM lipid trafficking, antagonizing effects of CKA-1 and SAMS-1 on lipids at ER membranes, and regulation of genes including *fat-7* through SBP-1. Other components of the pathway are omitted for clarity. Source data are provided as a Source Data file.

Despite increased PM leakiness resulting from reduced unsaturated lipids, the intestinal PM morphology and PM-targeted trafficking of proteins with GFP prenylation reporters appeared largely normal in *lpd-3* mutants (Supplementary Fig. 5b), suggesting specific roles of LPD-3 in lipid trafficking.

Together, these results support that LPD-3 promotes ER-to-PM trafficking of phospholipids, and thereby regulates SBP-1 nuclear abundance and expression of the *fat-7* gene.

## Organismic phenotype and phospholipid/Lecithin rescue of *C. elegans lpd-3* mutants

To determine the physiological role of LPD-3 at the organismic level, we characterized *lpd-3* mutant phenotypes in development and adult resilience to cold exposure. Compared with wild type, *lpd-3* mutants show developmental delay, reaching to the larval L4 stage more slowly (Fig. 4a). In adult stages, *lpd-3* mutants are strikingly sensitive to both cold shock (4 °C for 20 h) and short-term freezing shock (−20 °C for 25 min) (Fig. 4b, Supplementary Fig. 6a, b). As *lpd-3* mutants showed probable defective ER-to-PM trafficking of phospholipid, we sought to rescue such organismic phenotypes of *lpd-3* mutants by supplementation of phospholipids from various sources and individual constituents of phospholipids,

including choline, serine, ethanolamine and fatty acids (unsaturated oleic or saturated stearic acids). We found that phospholipids (from soy or egg yolks) or Lecithin (predominantly unsaturated PC-type glycerophospholipids), but not other compounds tested, fully rescued the developmental delay of *lpd-3* mutants (Fig. 4a, Supplementary Fig. 6a). Lecithin also markedly rescued adult survival to cold exposure in a dose-dependent manner (Fig. 4b–d). Additional defects of *lpd-3* mutants in fecundity, cold or freezing tolerance, locomotory behavior, and intestinal membrane integrity or permeability were also rescued by Lecithin (Fig. 4e, Supplementary Fig. 6b–d). These data further support the functional role of LPD-3 in ER-to-PM phospholipid trafficking (Fig. 4f) and demonstrate a compelling pharmacological means by using Lecithin or phospholipid compounds to rescue defects in *lpd-3* mutants.

## Conserved roles of LPD-3 family proteins in phospholipid trafficking and cold resilience

LPD-3 is the sole *C. elegans* orthologue of a highly evolutionarily conserved protein family including Tweek (*Drosophila*), Kiaa1109 (Zebrafish and mice) and KIAA1109 (Humans), recently renamed as BLTP1 (Supplementary Fig. 7a). To assess whether roles of LPD-3 in *C. elegans* are likely evolutionarily conserved in other organisms, we evaluated

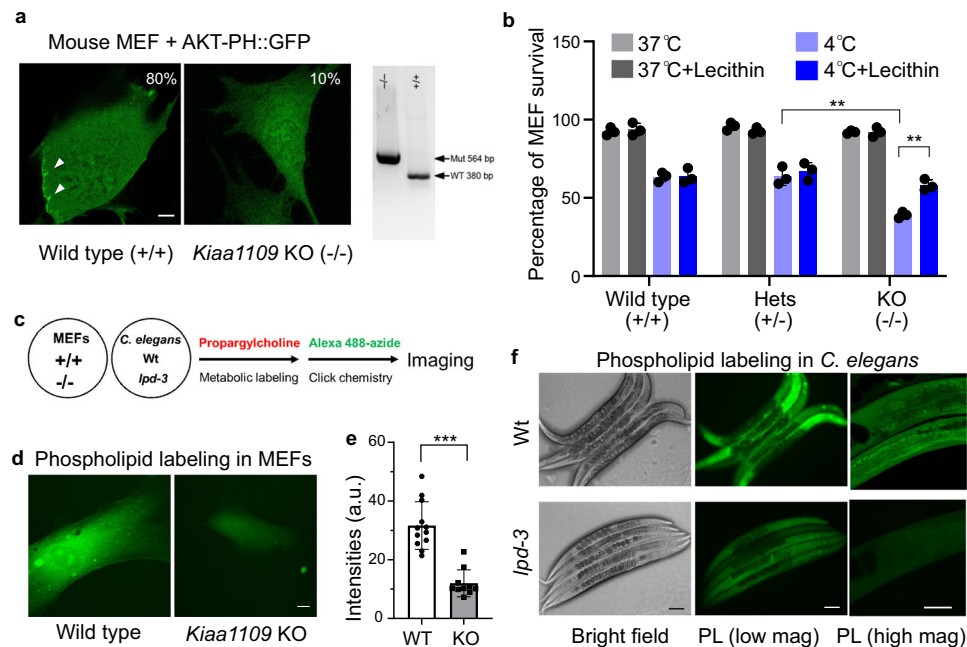

**Fig. 5 | Conserved roles of the LPD-3 homologue Kiaa1109 in MEFs.**
**a** Representative confocal fluorescence images showing enriched AKT-PH::GFP localization to ruffling membranes at cell periphery in wild-type (80%, $n = 20$) but less in *Kiaa1109* KO (10%, $n = 20$) MEFs. **b** Quantification of cell survival rates of MEFs based on SYTOX blue staining with indicated genotypes (wild type, heterozygous and homozygous knock-out of *Kiaa1109* in MEFs) after cold stress treatment. Values are means ± S.D with $**P < 0.01$ ($N = 3$ independent experiments, $n > 700$ cells per experiment; two-way ANOVA followed by multiple two-sided unpaired *t*-tests). **c** Schematic of metabolic phospholipid labeling using propargylcholine and click chemistry for visualization. **d** Representative confocal fluorescence images showing enriched propargylcholine signals with Alexa-488 fluorescence in wild-type and *Kiaa1109* KO MEFs. **e** Quantification of Alexa-488 fluorescence intensities (a.u., arbitrary unit) of propargylcholine-labelled phospholipids in wild-type ($n = 12$) and *Kiaa1109* KO ($n = 10$) MEFs. Values are means ± S.D with $***P < 0.001$ (two-sided unpaired *t*-test). **f** Representative bright field and fluorescence images showing typical enriched propargylcholine staining signals in wild type and *lpd-3(ok2138)* mutants. Scale bars, 50 μm. Source data are provided as a Source Data file.

the consequences of loss of *lpd-3* orthologues in both mammalian cells and Zebrafish. We derived and cultured mouse embryonic fibroblast (MEF) cells from *Kiaa1109*-deficient mouse embryos[50]. Transfection with AKT-PH::GFP reporters showed that PIP3 phospholipids as bound by AKT-PH::GFP were enriched at ruffling membranes of cell periphery in wild-type but not knock-out (KO) MEFs (Fig. 5a). *Kiaa1109* KO MEFs also exhibited higher sensitivity to cold stress, and this defect was rescued by supplementation with Lecithin (Fig. 5b). We also used a click chemistry-based method[51] to image phospholipids based on metabolic incorporation of the choline analog propargylcholine into phospholipids (Fig. 5c). We found that both *Kiaa1109* KO MEFs and *lpd-3* mutant *C. elegans* exhibited marked reduction of fluorophore-conjugated propargylcholine signals at the PM (Fig. 5d–f), while decreased overall abundance of such signals is consistent with reduced activity of SBP-1 in PC biosynthesis. In HEK293 human cell lines, we co-expressed two plasmids encoding AKT-PH::GFP and shRNA against *KIAA1109* and found that knockdown of *KIAA1109* led to reduced PM localization of AKT-PH::GFP (Supplementary Fig. 7b), as in MEFs and *C. elegans*. In addition, CRISPR/Cas9-mediated KO of *KIAA1109* in U937 cells also markedly increased cell death after 4 °C cold stress (Supplementary Fig. 7c). In Zebrafish, we used morpholino (MO) to knockdown *kiaa1109* and found that *kiaa1109* MO caused curved body and head defects as reported[18] (Supplementary Fig. 7d, e). Using a cold sensitivity assay[52], we found that knockdown of *kiaa1109* led to striking reduction of fish survival against cold stress (Supplementary Fig. 7f, g). Unlike *C. elegans*, Zebrafish embryos did not readily uptake exogenous lipids provided in their diets, thus precluding us from testing Lecithin effects in Zebrafish models. Nonetheless, the convergent phenotypes of phospholipid reporters, metabolic labeling and cold tolerance we have observed in *C. elegans*, Zebrafish, mouse and human cells strongly support the evolutionarily conserved roles of LPD-3 family proteins in promoting cellular phospholipid trafficking at

organelle membrane contact sites of ER and stress resilience to cold temperature.

## Discussion

Based on our integrated genetic, protein structural, cell biological and organismic phenotypic analyses, we propose a model for the role of LPD-3 in *C. elegans* (Fig. 4f). In this model, LPD-3 spans the ER and PM at localized membrane contact sites and acts as a megaprotein-based molecular bridge that mediates non-vesicular ER-to-PM trafficking of phospholipids. Such non-vesicular and rapid mode of lipid trafficking may be particularly important for meeting the demand of membrane expansion during development and for membrane fluidity adjustment during physiological adaptation to cold stress in adulthood. LPD-3-mediated proper flow of phospholipids from the ER to PM may also ensure appropriate levels of PC in ER membranes that, in turn, control the ER-to-nuclear translocation and abundance of SBP-1. In the nucleus, SBP-1 can regulate the expression of genes including *fat-7* and others involved in PC biosynthesis, lipogenesis, metabolic homeostasis, membrane property regulation and stress responses[21,24,25,35] (Fig. 4f).

We found striking rescue of all examined phenotypic defects of *lpd-3* mutants by exogenous supplementation of phospholipids or Lecithin. Ineffective rescue by phospholipid head group constituents or fatty acids indicates that phospholipid/Lecithin may act by incorporating into host membranes rather than providing simple nutritional support. Unlike eukaryote-derived polyunsaturated phospholipids, bacterial phospholipids from *E. Coli*, which *C. elegans* feeds on, contain mostly saturated lipids with little PC and PI[53] thus failing to support proper development and adult adaptation to cold in *lpd-3* mutants. Although our results suggest critical roles of LPD-3 in ER-to-PM lipid trafficking, LPD-3 may also localize and function at membrane contact sites formed by other ER-associated organelles[14]. Along with the

recently described family of VPS13/ATG2 lipid transporters, LPD-3 may represent an emerging class of lipid transporters that serve as molecular "highway bridges" critical for directed non-vesicular trafficking of lipids across different organelle membranes[12–16]. Although our data strongly support diverse phospholipids with unsaturated acyl chains as transported substrates by LPD-3, the precise substrate specificity and biophysical mechanisms of transport await further investigations.

Molecular functions of LPD-3 and its evolutionarily conserved orthologues have remained hitherto unknown. Its yeast orthologue Csf1 has been implicated in cold tolerance[54]. The *Drosophila* homologue Tweek regulates synaptic functions[55]. Mutations in the human orthologue KIAA1109/BLTP1 cause Alkuraya-Kucinskas syndrome, a neuro- and cardiovascular development disorder with no known medical treatment[17–19,56]. Loss of KIAA1109 also impairs phagocytosis of *L. pneumophila* by macrophages[57]. These divergent phenotypes may be underpinned by a unifying conserved role of this protein family in lipid trafficking. Rescue of *lpd-3* mutants in *C. elegans* by Lecithin suggests a similar route to treat the Alkuraya-Kucinskas syndrome. Potentially conserved roles of KIAA1109/BLTP1 and other mammalian homologues of LPD-3 in regulating lipid trafficking and lipogenesis also raise the possibility of targeting KIAA1109/BLTP1 in diverse lipid metabolic disorders, including fatty liver diseases and obesity.

## Methods

### *C. elegans*

*C. elegans* strains were maintained with laboratory standard procedures unless otherwise specified. The N2 Bristol strain was used as the reference wild type, and the polymorphic Hawaiian strain CB4856 was used for genetic linkage mapping and SNP analysis[58,59]. Forward genetic screens for *fat-7*p::GFP activating or suppressing mutants after ethyl methanesulfonate (EMS)-induced random mutagenesis were performed as described previously[3,60]. Approximately 25,000 haploid genomes were screened for *acdh-11* suppressors, yielding at least 18 independent mutants. Identification of mutations by whole-genome sequencing and complementation tests by crossing EMS mutants with *lpd-3(ok2138)* heterozygous males were used to determine *dma533* and *dma544* as alleles of *lpd-3*. Feeding RNAi was performed as previously described[61]. Transgenic strains were generated by germline transformation as described[62]. Transgenic constructs were co-injected (at 10–50 ng/µl) with dominant *unc-54p*::mCherry or GFP, and stable extrachromosomal lines of fluorescent animals were established. Genotypes of strains used are as follows: Chr. I: *lpd-3(dma533, 544, ok2138)*, *lpd-3(wy1770)*, knock-in allele of GFP11x7 to *lpd-3* C-terminus), Chr. III: *acdh-11(n5857)*; Chr. IV: *cka-1(dma550)*, Chr. V: *nIs590[fat-7::GFP]*, Chr. X: *sams-1(dma553, ok3033)*, *dmaEx647 [rpl-28p::lpd-3 Nt::mCherry; unc-54p::GFP]*, *dmaEx648 [ges-1p::mCherry::lpd-3 Ct; unc-54p::GFP]*, *epEx141 [sbp-1p::GFP::sbp-1 + rol-6(su1006)]*, *wyEx10612 [pDHC301 ges-1p:: mScarlet::CAAX (PM) 3 ng/ul, pDHC304 ges-1p:: GFP(1-10) 10 ng/ul, Pord-1::GFP 50 ng/ul]*, *wyEx10611 [pDHC302 ges-1p:: mScarlet::SP-12 (ER) 3 ng/ul, pDHC304 ges-1p:: GFP(1-10) 10 ng/ul, Pord-1::GFP 50 ng/ul]*, *wyEx10609 [pDHC303 ges-1p::ERSP::mScarlet::MAPPER (ER-PM contacts) 3 ng/ul, pDHC304 ges-1p:: GFP(1-10) 10 ng/ul, Pord-1::GFP 50 ng/ul]*, *pwIs503 [vha-6p::mans::GFP + Cbr-unc-119(+)]*, *hjIs73 [vha-6p::GFP::daf-22 + C. briggsae unc-119(+)]*, *xmSi01[mai-2p::mai-2::GFP]*; *epIs14 [sbp-1p::GFP + rol-6(su1006)]*, *pwIs890[Pvha-6::AKT(PH)::GFP]*, *jyIs13 [act-5p::GFP::act-5 + rol-6(su1006)]*, *zcIs4 [hsp-4::GFP]*.

### Structural prediction of LPD-3

The full-length LPD-3 was split into four fragments, each with ~1500 residues. Each fragment has ~500 overlapping residues with the neighboring fragments. Structure prediction of each fragment was generated by AlphaFold v2.0 (https://cryonet.ai/af2/) program[37]. Predicted structures were aligned using Chimera[63] based on the overlapping sequence. Then, the aligned structures of all fragments were merged in Coot[64] to obtain the full-length structure. C-terminal flexible loop was manually adjusted. The structural images were prepared in ChimeraX[65].

### Zebrafish

To investigate functions of *kiaa1109* in affecting cold resistance zebrafish larvae, the morpholino (MO) used to target zebrafish *kiaa1109* (E4I4) was obtained from Gene Tools[18]. Fertilized eggs of AB strain zebrafish were obtained as previously described[52]. The *kiaa1109* and ctrl MOs were dissolved in ultrapure water (5 ng/nL) and 1-2 nL MO solution was injected into each zebrafish egg at single cell stage using a PICO-LITER injector from WARNER. The injected embryos were incubated in E3 medium at 28 °C. The injected larvae with normal phenotype were selected and exposed to 10 °C at 96 hpf. After 24 h of cold exposure, the larvae were let to recover at 28 °C for 24 h. The fish were checked frequently and the dead ones were removed and counted. At the end of the experiment, the survived fish were classified as abnormal and normal as previously reported[52]. Photographs of the larvae before and after cold exposure were taken using a Zeiss stereomicroscope equipped with a color CCD camera. Body length of the larvae was measured by analyzing the photographs using AxioVision (v-4.8).

### Sample preparation for RNA sequencing, qRT-PCR and data analysis

N2 wild type and *lpd-3(ok2138)* animals were maintained at 20 °C and washed down from NGM plates using M9 solution and subjected to RNA extraction using TissueDisruptor and the RNeasy Mini Kit from Qiagen. RNA preparations were used for qRT-PCR or RNAseq. For qRT-PCR, reverse transcription was performed by SuperScript III, and quantitative PCR was performed using LightCycler Real-Time PCR Instruments. Relative mRNA levels were calculated by $\Delta\Delta CT$ method and normalized to actin. Primers for qRT-PCR: *act-3* (forward, tccatcatgaagtgcgacat; reverse, tagatcctccgatccagacg) and *fat-7* (forward, tgcgttttacgtagctggaa; reverse, caccaacggctacaactgtg). RNAseq library preparation and data analysis were performed as previously described[28]. Three biological replicates were included for each treatment. The cleaned RNAseq reads were mapped to the genome sequence of *C. elegans* using hisat2[66]. Abundance of genes was expressed as FPKM (Reads per kilobase per million mapped reads). Identification of differentially expressed genes was performed using the DESeq2 package[67].

### *C. elegans* cold and freezing stress and rescue by phospholipid/Lecithin

Animals were cultured under non-starved conditions for at least 4 generations before cold and freezing resilience assays. For cold resilience assay, bleach-synchronized L4 populations were kept at 4 °C for 20 h and then recovered for 24 h at 25 °C. For freezing resilience assay, bleach-synchronized L4 populations were kept at −20 °C for 45 min and then recovered for 24 h at 25 °C. For both cold and freezing experiments, NGM plates spread with equal agar thickness seeded with equal amounts of OP50 were used while cold and freezing temperature readings were monitored by thermometers to ensure minimal fluctuation. After cold or freezing shock, animals were moved to 25 °C for recovery and scored as dead if they showed no pumping and movement upon light touch with the body necrosis subsequently confirmed by light microscopy. For phospholipid and Lecithin rescue experiments, phospholipid (11145, Sigma-Aldrich), Lecithin (O3376-250, Fisher Chemical) or PC (P5394-10G, Sigma-Aldrich) was prepared as mixture by dissolving in M9 solution (from 1 to 20 mg/ml) and thorough vortexing. Phospholipids or Lecithin mixtures were then added (200 µl/60 cm plate) on NGM plates with pre-seeded OP50 and dried briefly before placing animals for cold or freezing tolerance assays.

### *C. elegans* development, fecundity and behavioral assays

To assay the developmental delay of *lpd-3* mutants, developmentally synchronized embryos from bleaching of gravid adult wild-type and *lpd-3* mutant hermaphrodites were plated on NGM plates and grown at 20 °C. After indicated duration (40, 45 and 50 h), percentages of animals reaching the L4 stage (with characteristic crescent vulvar structures) were quantified. To assay fecundity, single L4 worms were placed to control, phospholipid (20 mg/ml) or Lecithin (20 mg/ml) containing plates (prepared as above). After 72 h, the total numbers of progeny at all stages were scored. For locomotion behavioral assays, the average speed of worms was recorded for synchronized young adult hermaphrodite (24 h post L4) using the WormLab System (MBF Bioscience) based on the midpoint position of the worms[68]. Each experiment was repeated at least 3 times as independent biological replicates with more than 10 animals per group.

### Confocal and epifluorescence microscopic imaging

SPE confocal and epifluorescence compound microscopes (Leica) or LSM confocal microscope (Zeiss) were used to capture fluorescence images. Animals of different genotypes were randomly picked at the same young adult stage (24 hrs post L4) and treated with 1 mM Levamisole sodium Azide in M9 solution (31,742-250MG, Sigma-Aldrich), aligned on an 4% agar pad on slides for imaging. Identical setting and conditions were used to compare genotypes, experimental groups with control.

### Mammalian cell culture experiments

MEFs were derived from *Kiaa1109* mutant mice [B6N(Cg) −4932438A13Rik^tm1b(EUCOMM)Hmgu/J, Stock No.026878] generated by the Knockout Mouse Project (KOMP) at The Jackson Laboratory (Bar Harbor, Maine, USA) using embryonic stem cells provided by the International Knockout Mouse Consortium. *Kiaa1109*$^{-/-}$ embryos were obtained from interbreeding of heterozygotes. *Kiaa1109* mice were genotyped using the following PCR primers: wild-type allele (380 bp) forward GGG ATA TGG CAG AGA AGC TG, reverse AAA ACA ATT GGC TTA GAG ACT TCA; mutant allele forward CGG TCG CTA CCA TTA CCA GT, reverse GAC CAC ACA AAT CCC TTG GT. MEFs were cultured in DMEM (Thermo Fisher Scientific, MT-10−013-CV), supplemented with 10% FBS (Gemini Bio-Products, 900−208) and 1% penicillin/streptomycin and early passages (P2−P5) were used for reporter transfection, PC lipid labeling and cold resilience experiments. For phospholipid reporter transfection, MEFs were seeded at density of $4 \times 10^5$ cells/ml in 12-well plates containing glass cover slips and grown to 70−90% confluency. Mixture of DMEM, PloyJet reagent (Signagen Laboratories, MD, US) and CMVp::AKT-PH::GFP plasmids (Addgene) were prepared and added to wild-type and *Kiaa1109* KO MEF cultures, followed by imaging with fluorescence confocal microscopy after 48 h.

U937 cells (as suspension cultures) from ATCC were cultured in RPMI-40 (Gibco) medium supplemented with 10% heat-inactivated FBS (Hyclone), penicillin (10,000 I.U./mL), streptomycin (10,000 g/mL). HEK293T cells (as adherent cultures) were cultured in DMEM (Thermo Fisher Scientific, MT-10−013-CV), supplemented with 10% FBS (Gemini Bio-Products, 900−208) and 1% penicillin/streptomycin. Both cell lines were maintained in a humidified 5% CO2 incubator at 37 °C. U937 cells expressing lentiCas9-Blast were used to generate clonal lines of KIAA1109 KO with the sgRNAs targeting sequences GCCAGC-TACCCCCGAATAtgg and GTTGACATCTACTACTACAtgg. For cold stress experiments, parental control and KIAA1109 KO U937 cells were cold shocked (4 °C for 20 h) and assayed for cell death using CYTOX Green-based flow cytometry. For lipid reporter experiments, HEK293 cells were co-transfected with plasmids with AKT-PH::GFP and shRNA against *KIAA1109* (Sigma-Aldrich, TRCN0000263343 with 73% knockdown efficiency), incubated for 48 h and imaged by confocal microscopy for membrane localized GFP.

### Lipid metabolic labeling with propargylcholine

For PC lipid labelling in MEFs, *Kiaa1109*$^{+/+}$ and *Kiaa1109*$^{-/-}$ MEFs were incubated with propargylcholine (100 µM) in media for 24 h, fixed with 4% PFA in PBS for 5 min, reacted with 100 µM Alexa-488 Azide for 30 min. The cells were washed with PBS and imaged with fluorescence confocal microscopy. For PC lipid labelling in *C. elegans*, wild type and *lpd-3* mutants were cultured under non-starved conditions for at least 4 generations. L4-stage animals were incubated with 100 µM propargylcholine in OP50 culture for 24 h at 20 °C, fixed with 4% PFA in PBS for 5 min, reacted with 100 µM Alexa-488 Azide for 30 min, washed with PBS and imaged with fluorescence confocal microscopy.

### Statistics and reproducibility

Data were analyzed using GraphPad Prism 9.2.0 Software (Graphpad, San Diego, CA) and presented as means ± S.D. unless otherwise specified, with significance *P* values calculated by unpaired two-sided *t*-tests (comparisons between two groups), one-way or two-way ANOVA (comparisons across more than two groups) and adjusted with Bonferroni's corrections. Representative fluorescence images were shown for results repeated at least three times independently with similar results.

### Reporting summary

Further information on research design is available in the Nature Portfolio Reporting Summary linked to this article.

## Data availability

The RNAseq read datasets were deposited in NCBI SRA (Sequence Read Archive) under the BioProject accession PRJNA827259. All other data generated for this study are included in this paper, including those used for WormExp 2.0 (https://wormexp.zoologie.uni-kiel.de/wormexp/), Exon-Intron Graphic Maker (http://wormweb.org/exonintron), protein and nucleotide sequences (https://wormbase.org/) and AlphaFold v2.0 (https://cryonet.ai/af2/). Source data for all Figures are provided with this paper. Source data are provided with this paper.

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

## Acknowledgements

Some strains were provided by Drs. Barth Grant, Rosa E. Navarro González, and *Caenorhabditis* Genetics Center (CGC), which is funded by NIH Office of Research Infrastructure Programs (P40 OD010440). The work was supported by NIH grant 1R35GM139618, UCSF PBBR New Frontier Research (NFR) and the Packard Fellowship in Science and Engineering (D.K.M.).

## Author contributions

C.W., B.W., and D.K.M. designed, performed and analyzed most of the *C. elegans* experiments and wrote the paper. Yo.L. performed RNAseq bioinformatic analysis and Zebrafish experiments. T.P., F.O., R.G., and J.S. performed genetic mapping experiments and whole-genome sequencing analysis. J.Z. performed structural prediction and analysis. G.V. and J.M. contributed to lipid analysis. H.D. and K.S. contributed to CRISPR knock-in and confocal imaging analysis. M.B., S.M., Yu.L., W.L. contributed to the KIAA1109 sgRNA and KO cell experiments. D.K.M., C.Z. supervised the project.

## Competing interests

The authors declare no competing interests.
