## [Peer Review File · Nature Communications]

A conserved megaprotein-based molecular bridge critical for lipid trafficking and cold resilienceREVIEWER COMMENTS

Reviewer #1 (Remarks to the Author):

Trafficking of phospholipids from ER to PM is crucial for maintaining membrane integrity and altering membrane fluidity. In this manuscript, Dengke Ma and his colleagues identified a megaprotein LPD-3 that probably serves as a bridge to mediate non-vesicular ER-to-PM trafficking of phospholipids in *C. elegans*. More importantly, the roles of LPD-3/KIAA1109 family proteins are conserved across different species. In addition, they found that exogenous supplementation with lecithin rescued the defects caused by LPD-3/KIAA1109 deficiency, which might open up novel avenues to treat Alkuraya-Kucinskas syndrome. In general, although how LPD-3 proteins recognize and transport phospholipids is not yet fully understood, this study is nicely conducted and interesting. I have a few issues that need to be addressed.

1, In the structure prediction section, the authors separated LPD-3 into eight overlapping parts. Would you get the same structure with 4 segments or 6 segments?

2, Figure 2e is not clearly interpreted. From these images, it's hard to tell whether LPD-3 proteins localize to the ER membrane. In addition, they must provide images to show whether LPD-3 proteins localize to other organelles or not, which are missing in the current manuscript.

3, As shown in Figure 3f, knock-down of *lpd-3* clearly decreases the abundance of SBP-1::GFP in the cytosol and nuclei of intestine cells, thus the conclusion that LPD-3 regulates SBP-1 nuclear translocation is misleading. More experiments should be conducted to figure out how LPD-3 regulates SBP-1 abundance.

4, Does deficiency of LPD-3 alter the vesicular trafficking of lipids from ER to PM? If so, the authors should carefully revise their conclusions made in the manuscript.

Reviewer #2 (Remarks to the Author):

An important remaining question in cell biology is to understand how lipids are moved between cellular organelles. In this manuscript, the authors used powerful worm genetics to identify LPD-3/KIAA1109 as a possible "highway" for the transfer of lipids from ER to the plasma membrane. The authors provided strong genetic evidence, as well as cellular and physiological evidence. The structural prediction is also convincing. The disease relevance of KIAA1109 further elevates the potential impact of this work. This work is acceptable by NC. There are however some suggestions that will strengthen this work.

1. A major concern is the use of PIP sensors. PIPs are fairly minor lipids on the PM. Also, the loss of PIPs could result from reduced PI kinase activity on the plasma membrane, but not from defective transfer of PI. There are limited approaches to detect PC or PE or PI. However, there are good sensors for PS (see PMID: 33929485), which is highly abundant in the PM. The authors should examine PS in cells deficient in LPD-3/KIAA1109.

2. Also, a change in plasma membrane phospholipids often impact PM cholesterol (PMID: 33929485). This should also be checked. Moreover, supplementing or reducing cholesterol might help reduce cold sensitivity.

3. The accumulation of phospholipid in the ER could also result from increased synthesis. This possibility should be discussed, if not examined.

4. Data verifying knockdown or knockout efficiencies for all figures in this manuscript should be provided.

5. A proper introduction will also help the readers.

Reviewer #3 (Remarks to the Author):

Nature Communications manuscript NCOMMS-22-23331-T

Overview

This is an excellent manuscript from the Ma lab. Using forward genetics in *C. elegans*, they identified a previously uncharacterized protein that is required for the expression of a delta-9 desaturase, hence for homeoviscous adaptation for example to cold. The protein, LPD-3 in *C. elegans*, is very large (>4000 aa) and likely forms a sort of tunnel capable of facilitating lipid transport through its hydrophobic internal surface, likely allowing transfer of lipids from the ER to the plasma membrane. Data from human cell lines and zebrafish is also provided and suggests that the function of this protein is evolutionarily conserved. The paper is excellent and the main conclusions, i.e. that LPD-3 is essential for membrane homeostasis and likely facilitates transport of lipids from ER to plasma membrane, are well supported by the data. The importance and novelty of the findings, together with the evolutionarily conserved function of LPD-3 homologs, should guarantee a wide readership.

Major Comments

None. The paper is well structured and a pleasure to read, and its main conclusions well supported by the carefully presented experimental data. I found fascinating that the *lpd-3* mutation suppresses activation of *fat-7* by *sbp-1* likely because its ER is PC-rich and fluid.

Minor Comments

- Few details are provided concerning the forward genetics screen in *C. elegans*. How many haploid genomes were screened and how many mutants were isolated? Is *lpd-3* just one of many suppressors found? How many alleles of it were found (at least two are mentioned in the manuscript)?

- The question of lipid transport from ER to PM has been explored previously and an important theme is the presence of ER-PM contact points. I suggest mentioning this more clearly in the article and perhaps citing the following reference (and refs 4-9 therein): doi.org/10.1371/journal.pbio.2003864.

- The authors mentioned crossing the mCherry-tagged N-terminal LPD-3 reporter into a variety of strains with organelle-specific markers (lines 104-107) but do not show the results except for ER (Fig. 2e). Given that "data not shown" is usually not acceptable to many journals, it seems that the strains used and their sources should be specified, and the data presented.

Some questions remain unanswered, as is of course always the case.

- If the plasma membrane is depleted in phospholipids (as suggested from click chemistry experiments from Fig. 5, and also from Ext Data Fig. 7b), then what is structurally replacing them? Could the plasma membrane have an excess sphingolipids or other membrane lipids? Separate isolation of ER and plasma membranes from LPD-3 (or Tweek in fly, or KIAA1109 in humans) followed by lipidomics analysis would help confirm the plasma-membrane excess SFA/ER excess UFAs postulated by the authors, as well as inform on the actual composition of these membranes in the mutant. Indeed, a characterization of

membrane composition (i.e. lipidomics) in control and lpd-3 mutants would constitute a very important addition to the manuscript.

- Do exogenously supplied phospholipids become specifically incorporated in the plasma membrane and in this way suppress the lpd-3 mutant phenotypes? On the other hand, if the exogenously supplied phospholipids become inserted in many membranes shouldn't one expect that they would disrupt their functions in some way? Perhaps homeostatic mechanisms operate efficiently in the other membranes and the lpd-3 membrane defect is really restricted to the PM. A lipidomics analysis would again help in answering these questions.

- BLTP1, not KIAA1109, is now the approved symbol for the human homolog of LPD-3. BLTP1 is short for the full name "bridge-like lipid transfer protein family member 1" (https://www.genenames.org/data/gene-symbol-report/#!/hgnc_id/26953). BLTP1 is a member of the gene group "bridge-like lipid transfer protein family", which were elegantly described in Neuman et al (2022) (Ref 29 in the manuscript under review). Perhaps this info should be made clearer in the manuscript. The fact that lipid transport through the internal cavity of the tunnel formed by the homologs of LPD-3 is already somewhat established undermines slightly the novelty of the present manuscript.

Response to Reviewers

We thank reviewers for their constructive comments. Following their and editor's suggestions, we have revised the manuscript to address all the comments and remaining questions. We believe the revised manuscript has been further clarified and improved thanks to the reviewers. Below please see a point-to-point response:

REVIEWER COMMENTS

Reviewer #1 (Remarks to the Author):

Trafficking of phospholipids from ER to PM is crucial for maintaining membrane integrity and altering membrane fluidity. In this manuscript, Dengke Ma and his colleagues identified a megaprotein LPD-3 that probably serves as a bridge to mediate non-vesicular ER-to-PM trafficking of phospholipids in *C. elegans*. More importantly, the roles of LPD-3/KIAA1109 family proteins are conserved across different species. In addition, they found that exogenous supplementation with lecithin rescued the defects caused by LPD-3/KIAA1109 deficiency, which might open up novel avenues to treat Alkuraya-Kucinskas syndrome. In general, although how LPD-3 proteins recognize and transport phospholipids is not yet fully understood, this study is nicely conducted and interesting. I have a few issues that need to be addressed.

1, In the structure prediction section, the authors separated LPD-3 into eight overlapping parts. Would you get the same structure with 4 segments or 6 segments?

Thank you for the suggestion. Previously, eight overlapping fragments were used due to the protein size limitation of the alphafold prediction website. We have found an updated server (<https://cryonet.ai/af2/>) that can accommodate proteins of a larger size. We thus separated LPD-3 into 4 and 6 overlapping parts as suggested by the reviewer. Both predicted structures showed the same overall architecture as what we obtained from eight overlapping parts. The main structural elements were superimposed well with a slight difference in the curvature of the tube, and some differences were observed in the flexible loops. These results suggest that the structure prediction strategy is reasonably robust. In the revision, we updated the figure (2a-d, 4f) using the predicted structure based on 4 overlapping parts as it used the least number of fragments.

2, Figure 2e is not clearly interpreted. From these images, it's hard to tell whether LPD-3 proteins localize to the ER membrane. In addition, they must provide images to show whether LPD-3 proteins localize to other organelles or not, which are missing in the current manuscript.

We have now used the CRISPR/Cas9 technique to successfully generate a 7xGFP11 knock-in allele for LPD-3 at its C-terminus and included new data showing that endogenous LPD-3::GFP (complemented with 7xGFP1-10) clearly localizes to ER and specifically co-localizes with MAPPER, a marker for ER-PM membrane contact sites (Chang et al., 2013, PMID: 24183667), and does not co-localize with other organelle

markers (new Figure 2g, h, also shown below, and new Extended Data Fig. 4).

The original Figure 2e is based on single-plane confocal images taken from double transgenic animals expressing both the ER membrane marker *vha-6p::GFP::C34B2.10* (SP12) in the intestine and *rpl-28p::LPD-3Nt::mCherry* ubiquitously. SP12 (C34B2.10) is the 12-kD subunit of the signal peptidase on ER membranes and the integrated transgene *hjls14 [vha-6p::GFP::C34B2.10(SP12) + unc-119(+)] X* has been used in numerous publications (e.g. Xu et al., 2012, Lee et al., 2016 and Garcia, 2019) as an ER membrane marker and to study the diffusion behavior of ER membrane proteins in *C. elegans*. We observed that LPD-3Nt::mCherry signals form domains (arrow heads in Figure 2e) that localize within many hole-like regions of ER membranes perfectly surrounded by strong GFP::C34B2.10(SP12) signals. Since both green and red fluorescence images were taken from the same confocal plane, we interpret this striking pattern to mean that LPD-3Nt as labelled by mCherry localizes to specific sites of ER membranes. The new data we have collected from the knock-in allele, coupled with functional evidence for the role of LPD-3 in the ER-to-PM trafficking of lipids in subsequent studies, further support the notion that LPD-3 localizes at ER and anchors its N and C-terminus at ER and PM, respectively, thereby forming a bridge-like structure.

3, As shown in Figure 3f, knock-down of *lpd-3* clearly decreases the abundance of SBP-1::GFP in the cytosol and nuclei of intestine cells, thus the conclusion that LPD-3 regulates SBP-1 nuclear translocation is misleading. More experiments should be conducted to figure out how LPD-3 regulates SBP-1 abundance.

We have now changed the wording to more accurately indicate “SBP-1 abundance,” but do not exclude the possibility of nuclear translocation, since the weak and variable fluorescence of SBP-1::GFP resulting from the extrachromosomal array *epEx141 [sbp-1::GFP::SBP-1 + rol-6(su1006)]* precluded us from determining the ratio of cytosolic and nuclear SBP-1::GFP. Results from the laboratory of Anders Naar (Walker et al., 2011; PMID: 22035958) have previously shown that SBP-1 regulates *fat-7* expression through, at least in part, nuclear translocation and controlled by PC levels at ER membranes, and also showed altered abundance of SBP-1::GFP. This is consistent with diminished nuclear SBP-1::GFP in *lpd-3* mutants that accumulate PC at ER membranes because of defective ER-to-PM trafficking of phospholipids. Following this reviewer’s suggestion to figure out how LPD-3 regulates SBP-1 abundance, we tried to integrate *epEx141* for more uniform reporter expression, but did not obtain any integrants after multiple attempts.

We have also characterized another independently generated strain *wgls140 [sbp-1::TY1::EGFP::3xFLAG + unc-119(+)]* but did not observe any fluorescence. While we agree the detailed mechanism of SBP-1 regulation is interesting, we consider that it is beyond the scope of our current work and to be followed up in future studies.

4, Does deficiency of LPD-3 alter the vesicular trafficking of lipids from ER to PM? If so, the authors should carefully revise their conclusions made in the manuscript.

Vesicular trafficking of lipids is essential for animal development and viability. Indeed, RNAi or mutants of key *sec* genes involved in vesicular lipid trafficking pathways cause lethality. However, three *lpd-3* mutants as well as animals with RNAi against *lpd-3* we studied in this work are viable under standard laboratory conditions and only show striking vulnerabilities to severe cold stress. This argues against the idea that deficiency of LPD-3 alters the vesicular trafficking of lipids from ER to PM. In addition, the elongated tunnel-like structure of LPD-3 with subcellular localization at MAPPER-marked membrane contact sites bridging ER and PM is also inconsistent with vesicular trafficking of lipids.

Reviewer #2 (Remarks to the Author):

An important remaining question in cell biology is to understand how lipids are moved between cellular organelles. In this manuscript, the authors used powerful worm genetics to identify LPD-3/KIAA1109 as a possible "highway" for the transfer of lipids from ER to the plasma membrane. The authors provided strong genetic evidence, as well as cellular and physiological evidence. The structural prediction is also convincing. The disease relevance of KIAA1109 further elevates the potential impact of this work. This work is acceptable by NC. There are however some suggestions that will strengthen this work.

1. A major concern is the use of PIP sensors. PIPs are fairly minor lipids on the PM. Also, the loss of PIPs could result from reduced PI kinase activity on the plasma membrane, but not from defective transfer of PI. There are limited approaches to detect PC or PE or PI. However, there are good sensors for PS (see PMID: 33929485), which is highly abundant in the PM. The authors should examine PS in cells deficient in LPD-3/KIAA1109.

Although reduced PI kinase activity is a formal possibility that may explain our results, this would require a hitherto unknown mechanism by which deficiency of LPD-3 reduces PI kinase (*AGE-1/ PIKI-1/PPK-3* in *C. elegans*) activity. To our knowledge, there is no indication, evidence or plausible hypothetic scenario supporting how this may happen. In addition, *lpd-3* mutants do not mimic known phenotypes of *C. elegans* deficient in PI kinases. Instead, *lpd-3* mutants show numerous phenotypes (that can be rescued by exogenous phospholipids or Lecithin, Figure 4) consistent with a defective transfer of phospholipids (Figs 3, 4 and 5). Structure and subcellular localizations (new Fig. 2) of

LPD-3 also support that a defective transfer of PI, rather than reduced PI kinase activity, would be more plausible.

Nonetheless, we noted the caveats of using PIP sensors and decided to use a completely independent, click-chemistry-based, method (Figure 5f) to measure phosphatidylcholine (PC), the most abundant phospholipids in the cell. Both *C. elegans* and mammalian cells deficient in LPD-3/KIAA1109 showed markedly similar reduction of PC at the PM, again consistent with defective lipid transfer, rather than reduced PI kinase activity. Following this reviewer's suggestion, we have also generated transgenic strains of *C. elegans* expressing the genetically encoded reporter for phosphatidylserine (PS; GFP-evt2PH, Lact-C2-GFP; Uchida et al., 2011; Yeung et al., 2008), but did not observe apparent effects of *lpd-3* RNAi (which can suppress *fat-7::GFP* levels) on the PS reporter (see below representative 8 confocal images taken) as striking as those for PI/PC. While the absence of evidence is not the evidence for absence, it remains to be explored in future studies whether and how LPD-3-mediated lipid transport may exhibit lipid-species specificity.

2. Also, a change in plasma membrane phospholipids often impact PM cholesterol (PMID: 33929485). This should also be checked. Moreover, supplementing or reducing cholesterol might help reduce cold sensitivity.

We agree a change in plasma membrane phospholipids might impact PM cholesterol, which in turn affects membrane fluidity and cold sensitivity. However, *C. elegans* is a cholesterol auxotroph, requiring as little as 2.5 ng/mL to survive, and several studies have implicated roles of cholesterol primarily in hormonal signaling during development (e.g. Matyash et al., 2001, Kurzchalia et al., 2003). We agree that in-depth analysis of cholesterol is an interesting topic but beyond the scope of our current studies, which focus on the primary role of LPD-3 in phospholipids while cholesterol defects, if any, are likely secondary consequences. Technically, there is no reliable way to monitor the intracellular distribution of cholesterol in *C. elegans* as sufficiently sensitive and genetically encoded sensors have not been developed to monitor miniscule amounts of cholesterol in worm membranes compared to mammalian cell membranes.

3. The accumulation of phospholipid in the ER could also result from increased synthesis. This possibility should be discussed, if not examined.

Thank you for pointing this out. This is a possibility we did not consider in the paper. However, preliminary results from our ongoing collaboration with the McDonald lab at UT Southwestern indicate no increased overall abundance of phospholipids as measured from whole-animal extracts (see also below response to the reviewer #3). In vivo, we observed striking down-regulation of SBP-1, a master regulator of lipogenesis (Fig. 3f), and lipid droplets in *lpd-3* mutants (New Extended Data Fig. 5). These results, together

with subcellular localization (new Fig. 2) and functional roles of LPD-3 in ER-to-PM lipid trafficking (Fig. 3), argue against increased synthesis of phospholipids in *lpd-3* mutants.

4. Data verifying knockdown or knockout efficiencies for all figures in this manuscript should be provided.

We have now provided validation information on knockout efficiencies of the shRNAs against human *KIAA1109* and genotype confirmation of the knockout of *Kiaa1109* in mouse MEFs. In *C. elegans*, *lpd-3(ok2138)* was used as a homozygous 1913 bp genetic deletion, resulting in partial loss-of-function of *lpd-3* as *dma533* and *dma544* alleles did.

5. A proper introduction will also help the readers.

Thank you for the suggestion. We have now added an introduction.

Reviewer #3 (Remarks to the Author):

Nature Communications manuscript NCOMMS-22-23331-T

Overview

This is an excellent manuscript from the Ma lab. Using forward genetics in *C. elegans*, they identified a previously uncharacterized protein that is required for the expression of a delta-9 desaturase, hence for homeoviscous adaptation for example to cold. The protein, LPD-3 in *C. elegans*, is very large (>4000 aa) and likely forms a sort of tunnel capable of facilitating lipid transport through its hydrophobic internal surface, likely allowing transfer of lipids from the ER to the plasma membrane. Data from human cell lines and zebrafish is also provided and suggests that the function of this protein is evolutionarily conserved. The paper is excellent and the main conclusions, i.e. that LPD-3 is essential for membrane homeostasis and likely facilitates transport of lipids from ER to plasma membrane, are well supported by the data. The importance and novelty of the findings, together with the evolutionarily conserved function of LPD-3 homologs, should guarantee a wide readership.

Major Comments

None. The paper is well structured and a pleasure to read, and its main conclusions well supported by the carefully presented experimental data. I found fascinating that the *lpd-3* mutation suppresses activation of *fat-7* by *sbp-1* likely because its ER is PC-rich and fluid.

Minor Comments

- Few details are provided concerning the forward genetics screen in *C. elegans*. How many haploid genomes were screened and how many mutants were isolated? Is *lpd-3* just one of many suppressors found? How many alleles of it were found (at least two are mentioned in the manuscript)?

Thank you for your comments. We have now added such information to indicate approximately 25,000 haploid genomes screened. We have identified at least 18 *acdH-11* suppressors, two of which failed to complement *lpd-3(ok2138)* and are *lpd-3* mutations.

- The question of lipid transport from ER to PM has been explored previously and an important theme is the presence of ER-PM contact points. I suggest mentioning this more clearly in the article and perhaps citing the following reference (and refs 4-9 therein): doi.org/10.1371/journal.pbio.2003864.

Thank you. We have now added the relevant references.

- The authors mentioned crossing the mCherry-tagged N-terminal LPD-3 reporter into a variety of strains with organelle-specific markers (lines 104-107) but do not show the results except for ER (Fig. 2e). Given that “data not shown” is usually not acceptable to many journals, it seems that the strains used and their sources should be specified, and the data presented.

We have now added info about the strains. To address caveats of transgenic reporters and address where the endogenous LPD-3 may localize in the cell, we have now used CRISPR/Cas9 to generate a 7xGFP11 knock-in allele for LPD-3 and included new data showing that endogenous LPD-3::GFP (complemented with 7xGFP1-10) clearly localizes to ER and specifically co-localizes with MAPPER, a marker for ER-PM membrane contact sites (Chang et al., 2013, PMID: 24183667), and does not co-localize with other organelle markers (new Figure 2g, h, also shown below, new Extended Data Fig. 4).

Some questions remain unanswered, as is of course always the case.

- If the plasma membrane is depleted in phospholipids (as suggested from click chemistry experiments from Fig. 5, and also from Ext Data Fig. 7b), then what is structurally replacing them? Could the plasma membrane have an excess sphingolipids or other membrane lipids? Separate isolation of ER and plasma membranes from LPD-3 (or Tweek in fly, or KIAA1109 in humans) followed by lipidomics analysis would help confirm the plasma-membrane excess SFA/ER excess UFAs postulated by the authors, as well as inform on the actual composition of these membranes in the mutant. Indeed, a characterization of membrane composition (i.e. lipidomics) in control and *lpd-3* mutants would constitute a very important addition to the manuscript.

Thank you for the suggestion and these comments. We have initiated collaboration with the McDonald lab at UT Southwestern and now obtained preliminary results from whole-animal lipidomic profiling of wild-type and *lpd-3* mutants (N = 3 independent biological replicates). The new results show trends of reduced total levels of triglycerides in *lpd-3* mutants, consistent with down-regulation of SBP-1 and reduced abundance of lipid droplets. Though overall abundance of various phospholipids as measured from whole

animals are generally not affected in *lpd-3* mutants (slightly reduced PE may reflect feedback inhibition by accumulation of PC and its PEMT conversion from PE at ER), we did observe trends of an increase of HexCer-type sphingolipids (please see below Figure for reviewer). However, one should note the caveats of such whole-animal extract experiments to measure total lipid species without spatial resolution, which may not account for the differences of phospholipid species at ER versus PM in *lpd-3* mutants. We also consider that increased sample sizes (to reach sufficient statistic power for such lipidomic analysis) are required in future studies, coupled with independent imaging-based approaches once genetically encoded lipid sensors are developed and validated to determine with subcellular resolution which lipid species are affected by *lpd-3* *in vivo*.

- Do exogenously supplied phospholipids become specifically incorporated in the plasma membrane and in this way suppress the *lpd-3* mutant phenotypes? On the other hand, if the exogenously supplied phospholipids become inserted in many membranes shouldn't one expect that they would disrupt their functions in some way? Perhaps homeostatic mechanisms operate efficiently in the other membranes and the *lpd-3* membrane defect is really restricted to the PM. A lipidomics analysis would again help in answering these questions.

Yes, we also think exogenously supplied phospholipids become specifically incorporated in the plasma membrane and in this way suppress the *lpd-3* mutant phenotypes. Robust homeostatic mechanisms operate efficiently at ER membranes (see Walker et al., 2011; PMID: 22035958) that adjust lipid compositions via SBP-1 and *fat-7*.

- BLTP1, not KIAA1109, is now the approved symbol for the human homolog of LPD-3. BLTP1 is short for the full name "bridge-like lipid transfer protein family member 1" (https://www.genenames.org/data/gene-symbol-report/#!/hgnc_id/26953). BLTP1 is a member of the gene group "bridge-like lipid transfer protein family", which were elegantly described in Neuman et al (2022) (Ref 29 in the manuscript under review). Perhaps this info should be made clearer in the manuscript. The fact that lipid transport through the internal cavity of the tunnel formed by the homologs of LPD-3 is already somewhat established undermines slightly the novelty of the present manuscript.

Thank you for the comments. We were aware of this name change since we submitted the manuscript when the official name was still KIAA1109 in NCBI. We have now added the new name BLTP1 to indicate KIAA1109 we studied in the original manuscript.

REVIEWERS' COMMENTS

Reviewer #1 (Remarks to the Author):

The authors have fully addressed my questions. This is a nice piece of work. The paper is ready for publication. Congratulations!

Reviewer #2 (Remarks to the Author):

The authors have addressed my concerns and I think the paper is now suitable for publication.

Reviewer #3 (Remarks to the Author):

Well done, the authors have satisfactorily addressed my comments and the manuscript is still excellent. There is still a bit of a mystery about what may be replacing the phospholipids in the plasma membrane, but I have no further comments/questions.

REVIEWERS' COMMENTS

Reviewer #1 (Remarks to the Author):

The authors have fully addressed my questions. This is a nice piece of work. The paper is ready for publication. Congratulations!

Thank you.

Reviewer #2 (Remarks to the Author):

The authors have addressed my concerns and I think the paper is now suitable for publication.

Thank you.

Reviewer #3 (Remarks to the Author):

Well done, the authors have satisfactorily addressed my comments and the manuscript is still excellent. There is still a bit of a mystery about what may be replacing the phospholipids in the plasma membrane, but I have no further comments/questions.

Thank you.